# Lack of Endothelial α1AMPK Reverses the Vascular Protective Effects of Exercise by Causing eNOS Uncoupling

**DOI:** 10.3390/antiox10121974

**Published:** 2021-12-10

**Authors:** Thomas Jansen, Miroslava Kvandová, Isabella Schmal, Sanela Kalinovic, Paul Stamm, Marin Kuntic, Marc Foretz, Benoit Viollet, Andreas Daiber, Matthias Oelze, John F. Keaney, Thomas Münzel, Eberhard Schulz, Swenja Kröller-Schön

**Affiliations:** 1Department of Cardiology, University Medical Center Mainz, Johannes Gutenberg University, Langenbeckstr. 1, 55131 Mainz, Germany; miroslava.kvandova@gmail.com (M.K.); isabella.schmal@gmx.de (I.S.); sanelakalinovic@gmail.com (S.K.); paul.stamm@unimedizin-mainz.de (P.S.); marin.kuntic93@gmail.com (M.K.); daiber@uni-mainz.de (A.D.); matthias.oelze@unimedizin-mainz.de (M.O.); tmuenzel@uni-mainz.de (T.M.); swenja.kroeller-schoen@gmx.de (S.K.-S.); 2Centre National de la Recherche Scientifique (CNRS), Institut Cochin, INSERM, Université de Paris, 22, rue Mechain, F-75014 Paris, France; marc.foretz@inserm.fr (M.F.); benoit.viollet@inserm.fr (B.V.); 3German Center for Cardiovascular Research (DZHK), Partner Site Rhine-Main, Langenbeckstr. 1, 55131 Mainz, Germany; 4Division of Cardiovascular Medicine, UMass Medical School, 55 Lake Avenue North, Worcester, MA 01655, USA; jfkeaney@bwh.harvard.edu; 5Department of Cardiology, AKH Celle, 29223 Celle, Germany; eberhard.schulz@akh-celle.de

**Keywords:** α1AMPK, endothelial cells, exercise training, reactive oxygen species, endothelial dysfunction

## Abstract

Voluntary exercise training is an effective way to prevent cardiovascular disease, since it results in increased NO bioavailability and decreased reactive oxygen species (ROS) production. AMP-activated protein kinase (AMPK), especially its α1AMPK subunit, modulates ROS-dependent vascular homeostasis. Since endothelial cells play an important role in exercise-induced changes of vascular signaling, we examined the consequences of endothelial-specific α1AMPK deletion during voluntary exercise training. We generated a mouse strain with specific deletion of α1AMPK in endothelial cells (α1AMPK^flox/flox^ x TekCre^+^). While voluntary exercise training improved endothelial function in wild-type mice, it had deleterious effects in mice lacking endothelial α1AMPK indicated by elevated reactive oxygen species production (measured by dihydroethidum fluorescence and 3-nitrotyrosine staining), eNOS uncoupling and endothelial dysfunction. Importantly, the expression of the phagocytic NADPH oxidase isoform (NOX-2) was down-regulated by exercise in control mice, whereas it was up-regulated in exercising α1AMPK^flox/flox^ x TekCre^+^ animals. In addition, nitric oxide bioavailability was decreased and the antioxidant/protective nuclear factor erythroid 2-related factor 2 (Nrf-2) response via heme oxygenase 1 and uncoupling protein-2 (UCP-2) was impaired in exercising α1AMPK^flox/flox^ x TekCre^+^ mice. Our results demonstrate that endothelial α1AMPK is a critical component of the signaling events that enable vascular protection in response to exercise. Moreover, they identify endothelial α1AMPK as a master switch that determines whether the effects of exercise on the vasculature are protective or detrimental.

## 1. Introduction

Cardiovascular disease is associated with high mortality, despite broad treatment advances in this field [1,2]. Effective disease prevention would be capable of saving millions of patient lives and significantly reduce costs of our health system [3,4,5]. In this respect, endothelial dysfunction represents a common hallmark in the genesis of cardiovascular diseases such as atherosclerosis, hypertension and heart failure. Since endothelial dysfunction is known as one of the earliest events in the pathogenesis of cardiovascular disease, its prevention represents a major goal [6,7,8,9].

Among different forms of exercise training, “aerobic” voluntary exercise training is recognized as one of the most effective forms of disease prevention, since it promotes NO bioavailability and reduces oxidative stress in the vasculature. Both effects contribute to improved endothelial function and promote vascular health [10,11,12].

The endothelium contains important regulatory functions to maintain vascular homeostasis, including its ability to sense and transform physical forces through the blood flow into vascular signaling cascades. As it represents the natural border between blood flow and the vasculature, increased endothelial shear stress during exercise increases eNOS activity and thereby prevents vascular dysfunction and aging [13,14].

Several studies in mice and humans have shown that voluntary exercise training has many beneficial effects in the vascular system [15]. Moreover it is used as a non-pharmacological intervention for hypertension and heart failure [16]. Although many beneficial effects of exercise training have been discovered so far, molecular insights are scarce.

We and others have shown that exercise training also results in activation of important regulators of cell metabolism including AMP-activated protein kinase (AMPK), since it leads to energy consumption and a rise in the AMP/ATP ratio [17]. AMPK is known as a ubiquitously expressed metabolic master switch, which has important impact in vascular homeostasis and the control of reactive oxygen species production [18].

Previous work of our group demonstrated that the α1AMPK subunit that is predominantly expressed in vascular endothelial cells mediates the vascular-protective effects of exercise training, as its global deletion prevented eNOS up-regulation and suppressed antioxidant heme oxygenase-1 (HO-1) signaling [17]. Others have reported that genetic deletion of the α2AMPK subunit mediates the vascular-protective effects of exercise training, as beneficial effects of exercise were mostly lost in α2AMPK^−/−^ mice [19]. However, these observations were found in global α1AMPK knockout mice, while the role of α1AMPK in specific cell types remains unanswered. Since endothelial cells play a prominent role for vascular homeostasis, the current study examines the effects of voluntary exercise training in an endothelial-specific α1AMPK knockout mouse model (α1AMPK^flox/flox^ x TekCre^+^).

## 2. Materials and Methods

### 2.1. Animals

The experimental animals were cared for in agreement with the Guide for the Care and Use of Laboratory animals as declared by the U.S. National Institutes of Health. Furthermore, all experimental studies were performed with approval of the ethics committee of the University of Mainz (AZ.G13-1-032) and by the Institutional Animal Care and Use Committee (IACUC Guidelines).

For this study, mice with endothelial-specific deletion of α1AMPK (C57Bl/6 background; offspring of α1AMPK^flox/flox^ and TekCre^+^ breeding pair) and corresponding wild-type mice (TekCre^+^) were used. The offspring were genotyped and backcrossed with α1AMPK^flox/flox^ to obtain α1AMPK^flox/flox^ x TekCre^+^ [20,21]. The experiments were performed in 6–8 weeks. For voluntary exercise training (duration 7 weeks), mice were put into cages with a running wheel. A pedometer was adjusted to measure the running distance as described earlier [17]. A total of 72 male mice were used for this study.

A combination of Ketamine anesthesia and Xylazine analgesia, followed by cervical dislocation, was used for animal sacrifice. Next, sample collection of the aorta (thoracic and abdominal parts), heart, lungs and blood were performed for future examination as reported.

### 2.2. Materials

For immunoblotting analysis, the following antibodies were used: anti-AMPK and anti-3-nitrotyrosine purchased from Merck, KGaA (Darmstadt, Germany), anti-eNOS and anti-p-eNOS from BD biosciences (Heidelberg, Germany), anti-α-actinin and anti- ß-actin from Sigma-Aldrich, (Schnelldorf, Germany), and anti-Nrf-2 from Santa Cruz (Dallas, TX, USA). In addition, for immunohistochemical analysis, we used anti-endothelin-1 from ThermoFisher (Schwerte, Germany) and 3-nitrotyrosine antibody from Upstate Biotechnology (Waltham, MA, USA). Amplex Red Assay was applied to detect hydrogen peroxide (Molecular Probes, Eugene, OR, USA). Glyceryl-trinitrate (GTN; nitrolingual infusion solution, 1 mg/mL) from G. Pohl-Boskamp (Hohenlockstedt, Germany) was used for isometric tension studies. All other reagents were acquired from Sigma-Aldrich, Fluka, or Merck.

### 2.3. Determination of Vascular Reactivity and NO Levels

Isometric tension studies were used to determine vascular function in the aorta. Tension changes of the endothelium-intact aortic rings (thoracic aorta, 3 mm length) caused by cumulative addition of vasodilator acetylcholine (Ach; 1 × 10^−9^–5 × 10^−6^ M) and the endothelium-independent vasodilator glyceryl-trinitrate (GTN; 1 × 10^−9^–5 × 10^−5^ M) were analyzed by organ chambers upon pre-constriction by prostaglandin F2α as described previously [22].

Nitric oxide levels in the aortic rings were measured by using electron paramagnetic resonance (EPR)-based spin trapping with iron-diethyldithiocarbamate (Fe(DETC)_2_ colloid at 77 K using an X-band table-top spectrometer MS400 (Magnettech, Berlin, Germany) as described [23,24].

### 2.4. Vascular Reactive Oxygen Species (ROS) Production

Production of hydrogen peroxide was determined by HPLC-based Amplex Red/peroxidase assay of aortic tissue (4 mm rings, surrounding connective- and perivascular adipose tissue) [25].

Furthermore, dot blot analysis was done to determine levels of 3-nitrotyrosine-positive proteins as a marker of peroxynitrite-mediated oxidative damage in aortic homogenates using a specific anti-3-nitrotyrosine antibody (dil. 1:1000; Merck, KGaA, Darmstadt, Germany) as described before [26,27].

The in situ topographic detection of vascular superoxide production was performed by DHE staining (cytosolic superoxide) and mitoSOX staining (mitochondrial superoxide) of 6 µm thick cryosections of the aorta. In addition, fluorescence (green—aortic lamina autofluorescence; red—DHE/mitoSOX) was detected using a Zeiss Axiovert 40 CFL camera (Zeiss, Oberkochen, Germany) [28,29].

Moreover, the mitoSOX probe was applied for mitochondrial-specific production of ROS in isolated mitochondria from cardiac tissues [29,30] by mitoSOX/HPLC [31].

Immunohistochemical analysis was performed for determination of in situ topographically specific protein expression of 3-nitrotyrosine (Upstate Biotechnology, Waltham, MA, USA; dil. 1:1000) and endothelin-1 (ThermoFisher, Schwerte, Germany; dil. 1:1000) [26,27,32].

### 2.5. Isolation of Mouse Lung Endothelial Cells

Murine lung endothelial cells (MLECs) were isolated by digestion of the lungs using collagenase I (Worthington, LS 004216) followed by two-step separation of the cell suspension using CD31 MACS beads and ICAM Dynabeads according to instructions [32,33]. FACS sorting (FACSAria; BD, Heidelberg, Germany) of the MLEC stained with CD31 PE-CF594 (BD, Heidelberg, Germany), CD326 BV421 and CD45 PE-Cy7 (Biolegend, San Diego, CA, USA) and 7-AAD PerCP-Cy 5.5 (ThermoFisher, Frankfurt, Germany) was used for purification as described previously [34].

### 2.6. Reverse Transcription Real-Time PCR (qRT-PCR)

Quantitative real-time RT-PCR was used for the determination of mRNA expression in aortic tissue and endothelial cells. Total RNA was isolated using an RNeasy Fibrous Tissue Mini Kit (Qiagen, Hilden, Germany). The purity and concentration of isolated mRNA were determined by Eppendorf BioPhotometer (Eppendorf, Wesseling-Berzdorf Germany). Gene expression was determined by the QuantiTect^™^ Probe RT-PCR kit (Qiagen) using 50 ng of total RNA measured on StepOne (Applied Biosystems, Foster City, CA, USA) as previously described [35]. Taqman^®®^ Gene Expression assays for endothelial NO-synthase (eNOS, Mm00435204_m1), Nuclear factor E2 related factor-2 (NRF-2, Mm00477784_m1) heme oxygenase 1 (HO-1, Mm00516004_m1), NADPH oxidase 2 (NOX-2, Mm00432775_m1) and uncoupling protein 2 (UCP-2, Mm00627599_m1) were purchased as probe-and-primer sets (Applied Biosystems, Foster City, CA, USA). As the housekeeping gene, the TATA box binding protein (TBP; MM00446973_m1) was used. To determine mRNA quantification, the comparative ΔΔCt method was used and expressed as % of control.

### 2.7. Immunoblotting

Protein expression was determined by Western blot analysis in homogenates of aortic tissue or MLECs. Separation of the proteins was performed by SDS-PAGE electrophoresis followed by immunoblotting onto nitrocellulose membranes [36]. Antibodies—α1AMPK (1:500; 07-350, Merck KGaA, Darmstadt, Germany), α-actinin (1:2000; 2044, Sigma-Aldrich, Germany), ß-actin (1:2000; A5060, Sigma-Aldrich, Germany), eNOS (1:1000; 610297, BD, Heidelberg, Germany), p-eNOS Ser1177 (1:1000; 612393, BD, Heidelberg, Germany), 3-nitrotyrosin (1:200; 05-233, Merck KGaA, Darmstadt, Germany) and Nrf-2 (1:200; sc13032, Santa Cruz, Dallas, TX, USA) were used according to the manufacturer’s protocols. The positive bands were detected using species-matched secondary antibodies with covalently bound peroxidase (1:10,000; GAM-POX: PI-2000 and GAR-POX—PI-1000, Vector Laboratories, Burlingame, CA, USA) and an enhanced chemiluminescence detection kit—Pierce™ ECL Western Blotting Substrate kit (32106, Thermo Scientific), and quantification by a Chemilux chemiluminescence imager (CsX-1400M, Intas, Göttingen, Germany).

### 2.8. Serum Antioxidant Capacity

Serum antioxidant capacity was measured in the acetonitrile-deproteinized serum using 2,2-diphenyl-1-picryl-hydrazyl radical (DPPH) solution (50 μM) and determined by spectrophotometer at 517 nm every 10 min over 30 min. Serum antioxidant capacity was calculated as a decrease of absorbance due to antioxidant-mediated reduction of the DPPH radical [37].

### 2.9. Statistical Analysis

All results are expressed as the mean ± SD, and *p*-values < 0.05 were considered statistically significant. Statistical analysis was calculated using Prism for Windows, version 9, GraphPad Software Inc. For analysis, one-way ANOVA was used (with Bonferroni correction for comparison of multiple means; for comparison of body/heart weight, running distance, protein and mRNA expressions, aortic NO production and ROS productions) or two-way ANOVA with a repeated measurements approach (with Tukey correction for comparison of multiple means, for analysis of concentration-relaxation curves and serum antioxidant capacity).

## 3. Results

### 3.1. Mouse Model and Phenotype

Mice in the exercise group ran about 3000 m per day without significant differences among the groups (Figure 1A). Genetic deletion of α1AMPK in endothelial cells reduced protein expression by approximately 80% in α1AMPK^flox/flox^ x TekCre^+^ mice compared to wild-type controls (Figure 1B) and a >90% reduction in α1AMPK gene expression was shown in mouse lung endothelial cells (MLEC); see Appendix A. α1AMPK was up-regulated in response to 7 weeks of exercise training in MLEC. This up-regulation was absent in mice lacking endothelial α1AMPK (data not shown) (Figure 1C). While exercise reduced the body weight of wild-type mice, there was no significant change in the body weight of α1AMPKflox/flox x TekCre+ mice (Figure 1D–F). Since exercise training is known to improve endothelial function, we next performed isometric tension studies to evaluate the effects of endothelial α1AMPK deletion.

### 3.2. Exercise Training Induces Endothelial Dysfunction in Endothelial α1AMPK-Knockout Mice

Voluntary exercise training resulted in an improved endothelial function in wild-type mice. Surprisingly, mice with an endothelial α1AMPK deletion developed endothelial dysfunction in response to exercise (Figure 2A). Endothelial independent relaxation with Nitroglycerin (NTG) was not significantly affected, although it showed a similar pattern (Figure 2B). Since NO production and bioavailability are the strongest mediators of vascular relaxation, we measured vascular NO production by EPR. Aortic NO bioavailability increased as a consequence of voluntary exercise training in wild-type mice. This effect was blunted in α1AMPK^flox/flox^ x TekCre^+^ mice, despite no difference in baseline NO production (Figure 2C). To further investigate this unexpected finding, we next assessed endothelin-1 (ET-1) expression in the vasculature. ET-1 is a vasoactive peptide and is considered as a natural NO counterpart responsible for the progression of cardiovascular disease [38]. Interestingly, ET-1 expression was up-regulated in response to exercise training in the endothelium of mice lacking endothelial α1AMPK (Figure 2D).

### 3.3. Endothelial α1AMPK Deletion Results in Impaired eNOS Expression

In wild-type animals, voluntary exercise training resulted in a significant increase of eNOS gene and protein expression in endothelial cells and aortic tissue (Figure 3A–C). Mice lacking endothelial α1AMPK were not able to induce eNOS expression in response to exercise training. eNOS activation and function depends on its phosphorylation at Ser1177. Exercise training resulted in a not significant rise in aortic phospho-eNOS^Ser1177^ in wild-type animals, but this effect was blunted in α1AMPK^flox/flox^ x TekCre^+^ mice (Figure 3D). Since vascular function is dependent on the equilibrium of NO bioavailability and reactive oxygen species production, we next measured vascular oxidative burden.

### 3.4. Endothelial α1AMPK Disruption Aggravates Vascular ROS Production during Exercise

Exercise training resulted in a reduction of 3-nitrotyrosine formation in mouse lung endothelial cells of wild-type mice, indicating decreased peroxynitrite production. This is in sharp contrast to α1AMPK^flox/flox^ x TekCre^+^ mice, showing a strong increase in vascular peroxynitrite levels in response to exercise (Figure 4A). Immunohistochemistry confirmed these results, as we found significantly more 3-nitrotyrosine staining among all layers of the vascular wall (Figure 4B). Since peroxynitrite formation is a result of the reaction between NO and superoxide anions, we also assessed vascular reactive oxygen species production. As expected, vascular superoxide production measured by DHE was decreased in response to exercise training in wild-type mice, while again endothelial-specific α1AMPK deletion resulted in significantly elevated superoxide levels (Figure 4C). Aortic hydrogen peroxide levels were lower in wild-type mice performing exercise training, but no change was observed in our knockout animals (Figure 4D) in contrast to the observed rises in vascular peroxynitrite and superoxide production. Taken together, α1AMPK deletion in endothelial cells resulted in a pro-oxidative phenotype in response to exercise training.

### 3.5. Voluntary Exercise Training Induces eNOS Uncoupling in α1AMPK^flox/flox^ x TekCre^+^ Mice

Peroxynitrite is a strong oxidant and is considered a major mediator of eNOS uncoupling by the oxidation of tetrahydrobiopterin, S-glutathionylation and disruption of the dimeric eNOS enzyme [39,40]. Interestingly, we found evidence for eNOS uncoupling in the aorta of mice lacking endothelial α1AMPK undergoing exercise (Figure 5A). In wild-type animals, eNOS remained in a coupled state. These results support the contribution of uncoupled eNOS to exercise-induced vascular superoxide production in the setting of endothelial α1AMPK deficiency. Since the induction of eNOS uncoupling by peroxynitrite requires a primary ROS source, we next measured NOX-2 gene expression in mouse lung endothelial cells. While control mice exhibited a significant decrease in NOX-2 mRNA, mice with endothelial-specific deletion of α1AMPK displayed contrary results and showed an elevated NOX-2 expression (Figure 5B). A similar trend was present in aortic tissue (Figure 5C). As mitochondrial ROS may contribute to overall ROS production, we measured their abundance in aortic tissue by using mitoSOX staining. Exercise training decreased mitochondrial ROS levels in wild-type animals, but this effect was blunted in α1AMPK^flox/flox^ x TekCre^+^ mice performing exercise (Figure 5D). Additionally, we determined mitochondrial-specific ROS production using mitoSOX-based HPLC method in isolated heart mitochondria. Here we observed a similar effect as in aortic tissue. Exercise reduced mitochondrial ROS in wild-type mice, whereas in α1AMPK^flox/flox^ x TekCre^+^ animals, this protective effect of exercise was absent (Figure 5E).

### 3.6. Endothelial α1AMPK Deletion Results in Diminished Anti-Oxidant Capacities

Vascular oxidative stress might also be a consequence of an altered antioxidant defense. Therefore, we investigated the effects of exercise training on the antioxidant capacity using a DPP-radical-based assay. As expected, we found an improved antioxidant capacity in our knockout mice performing exercise training as envisaged by a faster decline in the DPP-radical (measured by decrease in absorption at 550 nm) (Figure 6A). Heme oxygenase 1 (HO-1) belongs to the most effective antioxidant enzymes, which are able to prevent ROS-mediated vascular damage [41]. We found a significant up-regulation of HO-1 mRNA in response to exercise training in wild-type controls but not in mice lacking endothelial α1AMPK (Figure 6B). Since HO-1 expression is Nrf-2 dependent, we found concordantly a decreased Nrf-2 expression in mouse lung endothelial cells and aortic tissue. As expected, Nrf-2 expression was increased in the aorta and mouse lung endothelial cells of wild-type animals undergoing exercise (Figure 6D–F). In addition, we investigated the gene expression of the antioxidant uncoupling protein 2 (UCP-2), which is located in the inner membrane of the mitochondria and protects cells from oxidative damage. In accordance with the previous finding, UCP-2 expression was up-regulated in response to exercise training in wild-type mice but decreased in mice lacking endothelial α1AMPK (Figure 6C).

## 4. Discussion

It is undoubtedly clear that voluntary exercise training is associated with beneficial effects on the cardiovascular system; therefore, it is even used in the treatment of cardiovascular diseases such as hypertension and diabetes mellitus in terms of life-style change [42,43,44]. An appreciable number of published data associate AMPK with the beneficial effects of physical exercise [45]. In the present study, we investigated the consequences of endothelial-specific α1AMPK deletion during voluntary exercise training in order to gain new mechanistic insights. Exercise training is known to promote vascular health by a significant reduction of oxidative stress and an improvement of endothelial function. Our most important and unexpected finding is that deletion of endothelial α1AMPK induces vascular oxidative stress in response to exercise training.

In our work, we demonstrate that endothelial α1AMPK has profound implications for the vascular benefits of exercise training, since its disruption reverses vascular protection and results in severe vascular oxidative damage and endothelial dysfunction. With our results, we demonstrate for the first time that voluntary aerobic exercise training may have detrimental effects on the cardiovascular system, depending on the integrity of endothelial α1AMPK. Our present results are in accordance with a former study that identified α1AMPK as a required element to mediate the protective vascular effects of exercise training. In contrast to the endothelial-specific α1AMPK deletion, however, a global deletion of α1AMPK did not result in endothelial damage in response to exercise. While mice with a global deletion of α1AMPK had no improvement of endothelial function in response to exercise training, animals with an endothelial-specific deletion of α1AMPK showed even an impairment of endothelial function [17]. Possible explanations for this discrepancy might relate to different genetic backgrounds (FVB vs. C57Bl/6J) or compensatory regulations.

Mechanistically, we found that eNOS uncoupling is a critical event that converts beneficial vascular effects of exercise into vascular damage. In the setting of endothelial α1AMPK deficiency, exercise training propagated eNOS-mediated superoxide production that was initiated by increased peroxynitrite formation indicated by elevated 3-nitrotyrosine levels. The question remains as to which events caused the initial ROS burst that triggered eNOS uncoupling via formation of “kindling radicals”. Several studies have shown that α1AMPK is implicated in ROS-mediated vascular homeostasis by interaction with specific NADPH oxidase isoforms [20,46,47,48,49,50]. For example, α1AMPK is known to limit angiotensin II-induced NOX-2 up-regulation [18]. Our actual results support the interaction between endothelial α1AMPK and NOX-2 in order to prevent ROS-mediated vascular injury, indicating that NOX-2 is an important AMPK-sensitive ROS source in the vasculature. Due to the high abundance of NOX-4 in the endothelium, we analyzed its mRNA expression previously but found no significant change in response to exercise, as opposed to NOX-2.

In the setting of ATII-induced endothelial dysfunction, endothelial α1AMPK deficiency is associated with enhanced immune cell infiltration in the vasculature and chemokine release [51]. Besides the endothelium, immune cells express their own NOX-2 to defeat bacterial pathogens. We did not investigate vascular inflammation in this present work, but future investigations will determine whether impaired endothelial function in α1AMPK knockout mice undergoing exercise might result in increased immune cell infiltration.

The key finding of the endothelial-specific α1AMPK-induced ROS production during exercise training modifies the paradigm that aerobic voluntary exercise training promotes unrestricted vascular health [52,53]. Our results unravel new mechanistic insights into the basic mechanisms of exercise physiology.

Loss of endothelial α1AMPK strongly enhances reactive oxygen species production during exercise training. Since ROS also originate from endothelial NOX-2, superoxide anions react with endothelial NO to stimulate peroxynitrite production, recognized as nitro-oxidative stress.

It is known that treatment of endothelial cells with peroxynitrite or its endogenous production in the setting of vascular disease results in eNOS uncoupling, a condition that converts eNOS from a NO-producing enzyme to a superoxide anions producing enzyme and perpetuates endothelial dysfunction [54,55,56]. Since the discovery of eNOS uncoupling, several factors have been identified that help explain the mechanistic background of this switch in enzymatic activity. These include the reduced bioavailability of tetrahydrobiopterin (BH_4_), which is an important coenzyme for the physiological function of eNOS and is considered to be critical [57,58]. BH_4_ levels were not measured in our study, but we found lower aortic GTP cyclohydrolase 1 (GTP-CH1) expression, the rate-limiting enzyme for BH_4_ synthesis, in endothelial α1AMPK knockout mice performing exercise (data not shown). Although we identified NOX-2 and uncoupled eNOS as crucial sources of superoxide production in our experiments, other superoxide-generating enzymes, e.g., xanthine oxidase or cytochrome P450, play important roles in ROS-mediated cardiovascular disease. The inhibition of xanthine oxidase has been shown to be associated with significantly reduced ROS production, which is closely related to endothelial dysfunction. Besides ROS production, effective anti-oxidative systems contribute to the sensitive equilibrium of redox-balance in the vasculature. Here, the superoxide dismutase (SOD) converts superoxide to hydrogen peroxide in order to lower oxidative stress. We measured the protein expression of SOD2 in aortic tissue and detected a not significant up-regulation in wild-type animals performing exercise, which was absent in our knockout animals (data not shown).

However, it is important to distinguish between different kinds of exercise training, since the mode or intensity of training determines the effects on the vasculature. For example, strenuous or high-intensity interval training (HIIT) has become very popular during the last few years, since it is known to have great effects on weight loss [59]. Compared to voluntary exercise, this kind of training is associated with vascular ROS production, due to a massive ATP requirement and production of lactic acid [60]. Another point of view is considering the disturbance of energetic homeostasis due to endothelial α1AMPK deletion as the initial event responsible for the increase of oxidative stress in the vasculature. This is in line with the significant increase in mitochondrial ROS production in our knockout mice performing exercise training. According to this, we found a decrease in UCP-2 expression in response to exercise training in the aortas of our knockout mice compared to the wild-type controls. The function of UCP-2 is still not completely understood, and it is still a matter of debate whether UCP-2 is up-regulated due to high ROS production and whether the feedback mechanism leads to its down-regulation or not [61]. Finally, UCP-2 is regarded as an antioxidant enzyme that is involved in the control of mitochondrial ROS production. It is also likely that UCP-2 function and expression is tissue specific, since many studies were performed in brown adipose tissue or in skeletal muscle. Here, we interpret the abrogated gene expression in response to exercise training as endothelial α1AMPK-dependent mitochondrial dysfunction, resulting in diminished antioxidant capacity.

In this respect, the potential impact of the individual fitness level has to be discussed. Regularly performed exercise training has beneficial effects, since it lowers heart rate, increases exercise performance and therefore has the capability to prolong lifetime [62]. Since the amount of α1AMPK in endothelial cells determines whether exercise training is beneficial or detrimental for vascular health, it needs to be questioned whether there is a high variation in α1AMPK expression levels among individuals. It would be interesting to measure endothelial α1AMPK expression in individuals to investigate the correlation between fitness performance and endothelial function in response to exercise training. Examining α1AMPK expression in endothelial progenitor cells of individuals performing exercise training could be a strategy to investigate this aspect.

## 5. Conclusions

In summary, exercise training is known to have beneficial effects in the vasculature, as it reduces the rate of cardiovascular complications and decelerates the progression of atherosclerotic disease. Our study demonstrates a unique scenario with unexpected harmful consequences of exercise training due to endothelial-specific α1AMPK deletion. So far, AMPK was known as a “metabolic master switch” that determines the fate of a single cell during energy deprivation. Our results extend this principle to the whole organism since the lack of endothelial α1AMPK turns exercise from a vascular protective to a vascular damaging principle.

## Figures and Tables

**Figure 1 antioxidants-10-01974-f001:**
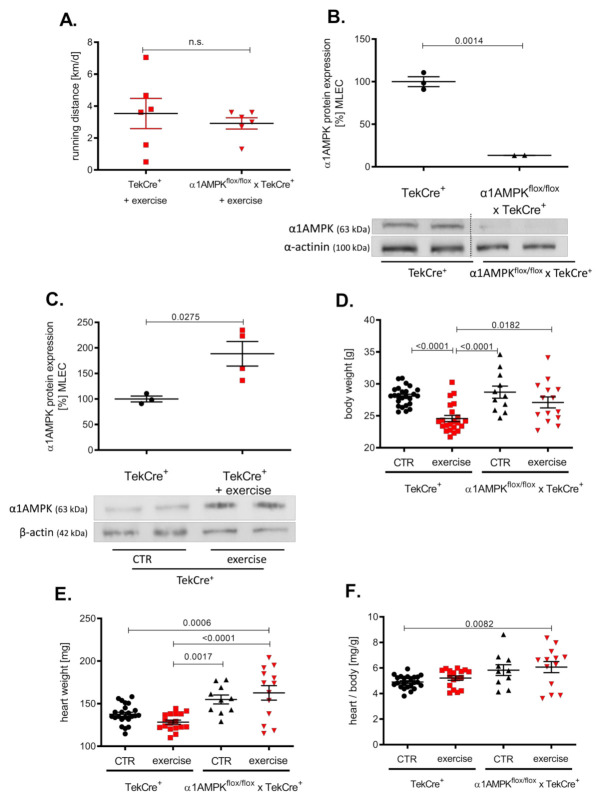
Effects of exercise training on α1AMPK expression in the endothelium. Voluntary exercise training was performed on running wheels supplied in mouse cages, which were connected to a counter to measure running distance. Running distances were shown in km/d (**A**). Western blot experiments present the consequence of exercise training on α1AMPK protein expression in isolated mouse lung endothelial cells (MLEC) in α1AMPK^fl/fl^ x TekCre^+^ and corresponding wild type controls (TekCre^+^). Representative western blot of 6 independent experiments is shown (**B**). Effects of voluntary exercise training on body and heart weight were shown for every group (**C**–**F**). Data are shown as mean ± SEM from 11–25 animals per group.

**Figure 2 antioxidants-10-01974-f002:**
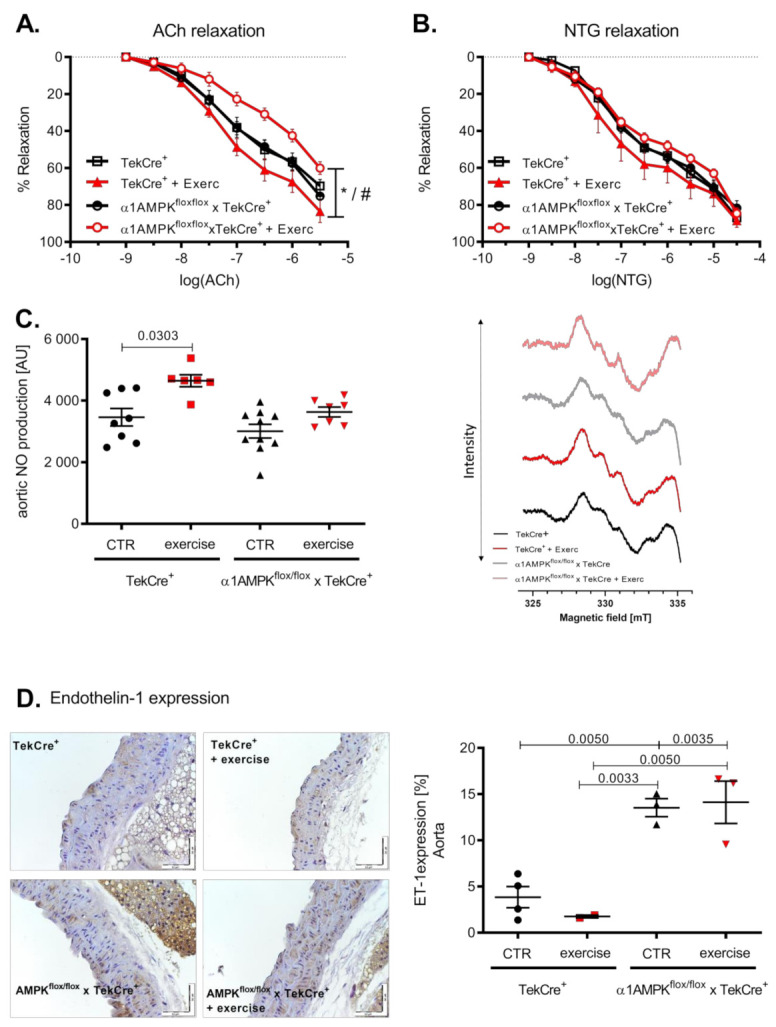
Mice with endothelial α1AMPK develop endothelial dysfunction in response to exercise training. Vascular relaxation in aortic rings was investigated in organ chamber experiments. Endothelial dependent relaxation with cumulative doses of Acetylcholine and endothelial independent relaxation with cumulative doses of Nitroglycerin are shown (**A**,**B**). Data are shown as mean ± SEM from 8–16 animals per group. Aortic NO production was assessed using EPR in repetitive measurements (**C**). Representative EPR traces are shown. Effects of exercise training on aortic endothelin-1 expression are presented with immunohistochemistry and the respective densitometry (**D**). Data are shown as mean ± SEM from 3–10 animals per group. * means *p* < 0.05 vs. TekCre^+^. # means *p* < 0.05 vs. α1AMPK^fl/fl^ x TekCre^+^ + Exercise.

**Figure 3 antioxidants-10-01974-f003:**
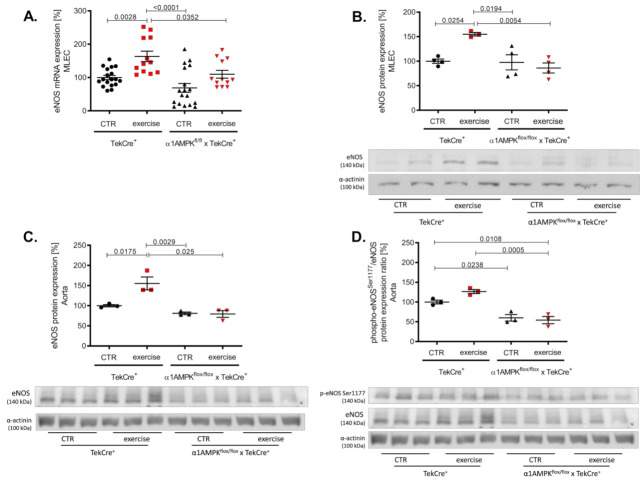
Deletion of endothelial α1AMPK diminishes eNOS expression. eNOS gene expression in response to exercise training was measured with real-time PCR in isolated mouse lung endothelial cells (**A**). Abrogated protein expression of eNOS in α1AMPK^fl/fl^ x TekCre^+^ mice after exercise training in MLEC was determined in western blot experiments (**B**). Representative western blot is shown. Immunoblots of eNOS protein expression in the aortic tissue (**C**) phospho-eNOS^Ser1177^/eNOS protein expression ratio (**D**) during exercise training are shown. Data represent as mean ± SEM. with pooled samples of 12-18 animals in each group.

**Figure 4 antioxidants-10-01974-f004:**
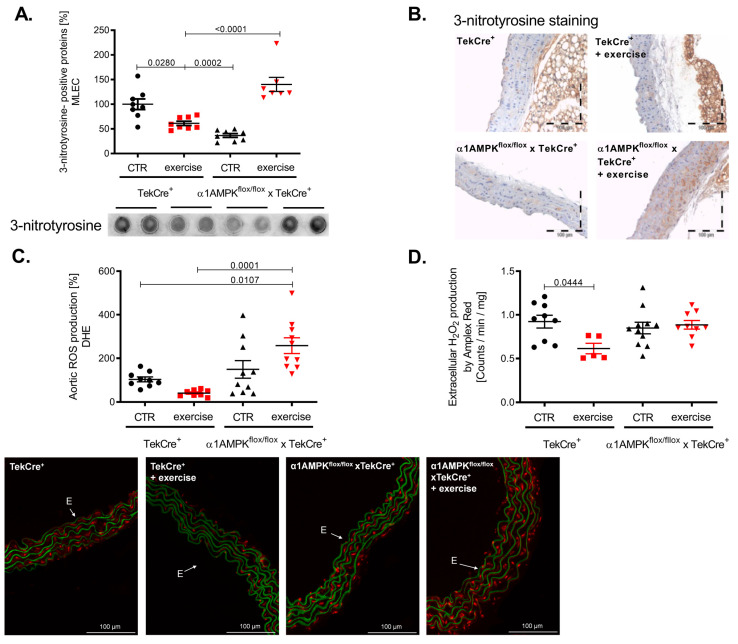
Reactive oxygen species production in response to exercise training is aggravated in endothelial specific α1AMPK knockout mice. Effects of exercise training on 3-Nitrotyrosin formation in MLEC were determined by dot blot analysis (**A**). Immunohistochemistry staining using the 3-nitrotyrosin antibody was performed in sections of aortic tissue (**B**). Superoxide production in the vasculature was measured by in situ topographic DHE- staining in aortic cryosections (**C**). Representative pictures are shown. Production of hydrogen peroxide in the aortic tissue was determined with Amplex Red assay (**D**). Data are mean ± SEM from 5–11 mice per group.

**Figure 5 antioxidants-10-01974-f005:**
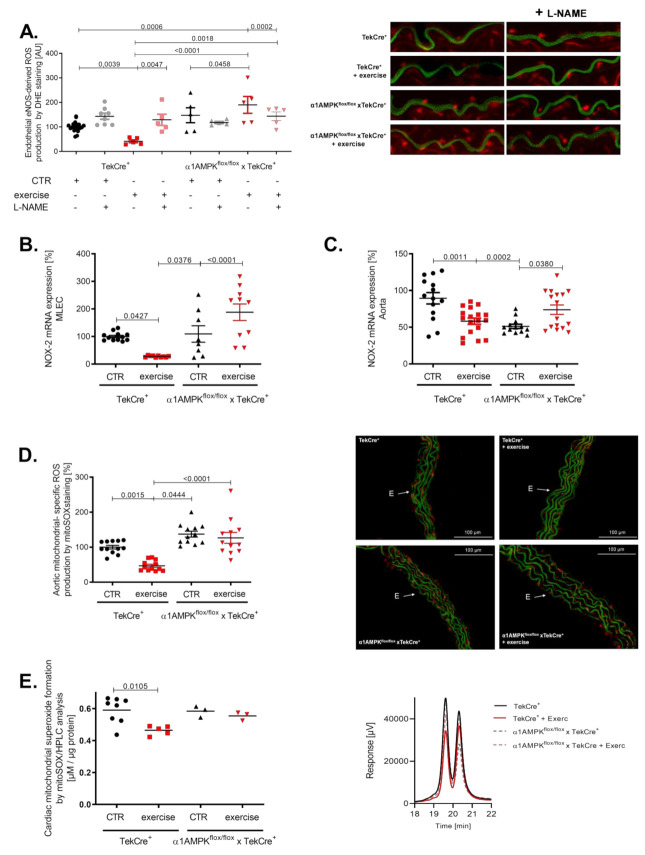
Exercise training enhances ROS production and eNOS uncoupling in endothelial α1AMPK knockout mice. eNOS uncoupling in α1AMPK^fl/fl^ x TekCre^+^ mice was shown in aortic cryosections using DHE staining and the NO-synthase inhibitor L-NAME. This results in exaggerated ROS production, aggravated in response to exercise in α1AMPK^fl/fl^ x TekCre^+^ (**A**). NOX-2 mRNA expression in mouse lung endothelial cells (**B**) and aortic tissue (**C**) was determined with real-time PCR. Mitochondrial reactive oxygen species production were measured by using mitoSox staining in prepared cryosections of aortic tissue (**D**). Additionally we measured mitochondrial- specific reactive oxygen species production using mitoSox—HPLC based method, n = 3–9 (**E**). Representative chromatograms are presented for each group. Data are mean ± SEM from 3–20 mice per group or pooled samples from at least two mice.

**Figure 6 antioxidants-10-01974-f006:**
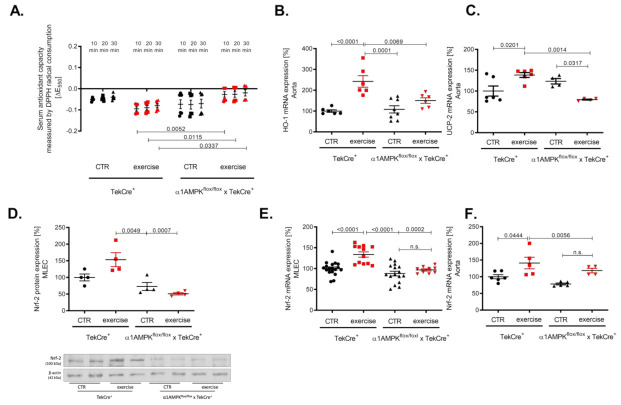
Endothelial α1AMPK deletion results in diminished antioxidant capacity. Antioxidant capacity was determined with a DPPH assay in serum of mice performing exercise (**A**). Results display repetitive measurements of five independent experiments. Aortic gene expression of heme oxygenase 1 (**B**), Nrf-2 mRNA expression (**F**) and UCP-2 gene expression (**C**) were measured with quantitative real-time PCR. Representative western blot with densitometric analysis of Nrf-2 protein expression in isolated mouse lung endothelial cells of all experimental animal groups are shown (**D**). Each lane represents a pooled sample of at least three animals. Nrf-2 mRNA expression was additionally measured in mouse lung endothelial cells (**E**). Data are mean ± SEM from 4–18 mice per group.

## Data Availability

Data is contained within the article and Appendix A.

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
