# Peer review of "Lack of Endothelial α1AMPK Reverses the Vascular Protective Effects of Exercise by Causing eNOS Uncoupling"

_antioxidants, 2021, doi:10.3390/antiox10121974_

Round 1

Reviewer 1 Report

I congratulate you for a really precious work in its design, in the methods and in the results obtained. I have only observed what I think is a small error in the title of figure 2 In Figure 2, where it says "Mice with endothelial ...." it should say "Mice with delection endothelial ..."

Author Response

Response to Reviewer No.1

Comments and Suggestions for Authors

I congratulate you for a really precious work in its design, in the methods and in the results obtained. I have only observed what I think is a small error in the title of figure 2 In Figure 2, where it says "Mice with endothelial ...." it should say "Mice with delection endothelial ..."

Answer:

Thank you kindly for the acknowledgement and recognition of our work. We corrected the title of figure 2 properly.

Reviewer 2 Report

In this study, the authors report that the ablation of α1AMPK in the endothelium led to the loss of vascular protective role of endothelial cells after voluntary exercise such as endothelial dysfunction, increase of ROS levels and eNOS uncoupling. They also report that superoxide producing enzyme (NOX2) was up-regulated and ROS scavenging system was down-regulated. This study has significance in that AMPK is involved in reducing ROS elvels, especially superoxide and regulating NO levels in the endothelial cells. However, there are major concerns that need to be answered before publication.

Major concerns:

  1. In Figure 1B, α1AMPK-knockout EC still have a lot of proteins expressed. They report only 60% of proteins were reduced by knockout (generally, more than 90% of proteins should be reduced by gene knockout). It should be explained why many of target proteins were expressed in the knockout cells and these amounts of protein is sufficient to give such effects to the cell as described in the manuscript.
  2. Westerns in Figure 3B, C and D are not clear. There are multiple bands near the target band and the backgrounds are high especially in Fig 3C and D. They should be improved before scanning the images. In the text on lines 231-232, the authors interpreted Figure 3D as stating “Exercise training resulted in a significant rise in aortic phosphor-eNOSSer1177 in wild type animals, ~”, however, it is hard to tell phosphor-eNOSser1177 is increased by exercise in the wild type, because there is no significance value marked between control and exercise group, and moreover, the western images do not show any distinct increase of phosphorylation of α1AMPK in the exercise group of wild type. It does not seem to be that the ratio of phosphor-/non phosphor-α1AMPK was decreased in the knockout mice compared to that of wild type.
  3. In general, NOX-2 is mainly expressed in phagocytic cells, and NOX-4 is predominantly expressed in vascular endothelial cells (see the paper, PMID: 15706079). Therefore, the expression of NOX4 should be examined and discussed.

Minor points:

  1. In Figure 4C, it should be explained what the green and red signals designate.
  2. In Figure 5A, and 5D, what do the green and red signals mean? What do the arrows indicate?
  3. Please state the full name of the abbreviation of NTG at the first place where it appears.

Author Response

Response to Reviewer No.2

Major concerns:

  1. In Figure 1B, α1AMPK-knockout EC still have a lot of proteins expressed. They report only 60% of proteins were reduced by knockout (generally, more than 90% of proteins should be reduced by gene knockout). It should be explained why many of target proteins were expressed in the knockout cells and these amounts of protein is sufficient to give such effects to the cell as described in the manuscript.

Answer:

Thank you very much for highlighting this important point. The relatively low degree of gene suppression at the protein level might be due to necessary purification process as we used purified mouse lung endothelial cells. A possible contamination with other cell types might lead to the observed residual amount of a1AMPK, as it is an ubiquitous expressed enzyme. By immunhistochemistry we were able to show a complete reduction of endothelial a1AMPK staining in a previous work using the same mouse model (please see “Endothelial α1AMPK modulates angiotensin II-mediated vascular inflammation and dysfunction” by Kröller-Schön et al.; Basic Res Cardiol. 2019 Jan 14;114(2):8. doi: 10.1007/s00395-019-0717-2.

We added this background information in the corresponding result part.     

  1. Westerns in Figure 3B, C and D are not clear. There are multiple bands near the target band and the backgrounds are high especially in Fig 3C and D. They should be improved before scanning the images. In the text on lines 231-232, the authors interpreted Figure 3D as stating “Exercise training resulted in a significant rise in aortic phosphor-eNOSSer1177in wild type animals, ~”, however, it is hard to tell phosphor-eNOSser1177is increased by exercise in the wild type, because there is no significance value marked between control and exercise group, and moreover, the western images do not show any distinct increase of phosphorylation of α1AMPK in the exercise group of wild type. It does not seem to be that the ratio of phosphor-/non phosphor-α1AMPK was decreased in the knockout mice compared to that of wild type.

Answer:

We understand your concerns regarding the interpretation of the shown western blot results. It is important to note that for the purpose of densitometric analysis we clearly identified the target band using the appropriate marker.

After recalculation of the densitometry results at different exposure times, there is no significant increase in the ratio of p-eNOS expression / eNOS expression in response to exercise training. Therefore, we corrected the figure and figure legend. The corresponding part in the manuscript was corrected as “no significant rise”. We apologize for that misinterpretation.

  1. In general, NOX-2 is mainly expressed in phagocytic cells, and NOX-4 is predominantly expressed in vascular endothelial cells (see the paper, PMID: 15706079). Therefore, the expression of NOX4 should be examined and discussed.

 Answer:

This is a very important issue. Due to the high abundance of Nox4 in the endothelium, we analyzed its mRNA expression previously, but found no significant change in response to exercise, as opposed to NOX-2. We added this information to the discussion of the manuscript.

Minor points:

  1. In Figure 4C, it should be explained what the green and red signals designate.

Answer:

First of all, we want to apologize for that lack of information. We included an explanation to the figure legend.

”Representative pictures are shown; aortic lamina autofluorescence (green), superoxide (red), E= endothelium.”

  1. In Figure 5A, and 5D, what do the green and red signals mean? What do the arrows indicate?

Answer:

The required information was added to the figure legend. The arrows were used for orientation purposes and to highlight the endothelium. Thank you very much for this important comment.

5A: ” Representative pictures are shown, aortic lamina autofluorescence (green), superoxide (red). ”

5D: ” Representative pictures are shown; aortic lamina autofluorescence (green), mitochondrial-specific superoxide (red), E= endothelium. ”

  1. Please state the full name of the abbreviation of NTG at the first place where it appears

Answer:

We apologize for that and stated the full name of Nitroglycerin (NTG) in the first place.

Reviewer 3 Report

General Comments:

This work investigated the role of endothelial α1AMPK in preventing cardiovascular disease via voluntary exercise training. A mouse strain with specific deletion of α1AMPK in endothelial cells was generated. Compared to the wild-type mice, instead of improving endothelial function, the voluntary exercise training elevated reactive oxygen species production, induced eNOS uncoupling and endothelial dysfunction in α1AMPK deleted mice. The results demonstrate that endothelial α1AMPK plays a critical role in the signaling events that determine if exercise is beneficial or harmful to the vascular function. Overall, it is a well-written manuscript. However, several concerns need to be addressed, please see below.

Specific Comments:

Introduction

  1. What is the difference between global α1AMPK knockout and endothelial specific α1AMPK knockout mouse models?

Materials and Methods

  1. It was indicated that 72 male mice were used in the study. How about female mice? Any specific reason for only using male mice?

Results

  1. Fig. 1B should be before Fig. 1A. In Line 192-193, It says “The deletion of α1AMPK in endothelial cells did not affect body weight or heart weight in animals performing exercise training (Figure 1C-E)”. However, Fig. 1C shows that exercise reduces the body weight of the wild-type mice, but no significant change in the body weight of the knockout mice. Fig. 1D also shows there is difference in the heart weight between exercised wild-type and knockout mice.

  1. Fig. 1B shows that genetic deletion of α1AMPK in endothelial cells reduced protein expression by 60% in α1AMPKflox/flox x TekCre+ mice compared to wild-type controls in mouse lung endothelial cells. How about in aortic endothelial cells?

  1. In Fig. 2 caption, Mice “with” not “without”?

  1. In Fig. 3A, it looks like that there is a significant difference between the CTR and exercise also for the knockout mice. Please check.

  1. Why there is no data for eNOS mRNA for aortic endothelial cells? And no data for the p-eNOS/eNOS for lung endothelial cells?

  1. In Fig. 4C images, are the red spots indicating the ROS production-DHE? More detailed description is suggested for the caption.

  1. More detailed description is necessary for the caption in Fig. 5.

Discussion

  1. More detailed description should be given for the discrepancy in vascular function between the global and endothelial specific α1AMPK knockout mice.

Author Response

Response to Reviewer No.3

Introduction

  1. What is the difference between global α1AMPK knockout and endothelial specific α1AMPK knockout mouse models?

Answer:  

The global α1AMPK knockout mice were bred on another genetic background compared to the tissue specific α1AMPK knockout. The “FVB strain” was used in global α1AMPK knockout mice, whereas we used TecCre mice for the tissue specific deletion of α1AMPK, which were based on a B6J background.

Functionally, the FVB strain displays a slightly shifted vascular relaxation curve compared to the TecCre wildtype control. Overall, global α1AMPK deletion just restored vascular ROS production in response to exercise, whereas endothelial specific deletion aggravated oxidative stress in the vasculature. 

Materials and Methods

  1. It was indicated that 72 male mice were used in the study. How about female mice? Any specific reason for only using male mice?

 Answer:

We only used male mice in our experiments to avoid possible endocrine side effects. Estradiol is a potent AMPK activator (see Estradiol-mediated endothelial nitric oxide synthase association with heat shock protein 90 requires adenosine monophosphate-dependent protein kinase by Schulz-E et al., Circulation. 2005 Jun 28;111(25):3473-80. doi: 10.1161/CIRCULATIONAHA.105.546812. Epub 2005 Jun 20.) and varying estradiol levels in female mice might have masked the observed effects of exercise.

Results 

  1. Fig. 1B should be before Fig. 1A. In Line 192-193, It says “The deletion of α1AMPK in endothelial cells did not affect body weight or heart weight in animals performing exercise training (Figure 1C-E)”. However, Fig. 1C shows that exercise reduces the body weight of the wild-type mice, but no significant change in the body weight of the knockout mice. Fig. 1D also shows there is difference in the heart weight between exercised wild-type and knockout mice.

 Answer:

We apologize for this misinterpretation. We corrected the mentioned part of the manuscript in the following way “While exercise reduced the body weight of wild-type mice, there was no significant change in the body weight of α1AMPKflox/flox x TekCre+ mice”.

  1. Fig. 1B shows that genetic deletion of α1AMPK in endothelial cells reduced protein expression by 60% in α1AMPKflox/flox x TekCre+ mice compared to wild-type controls in mouse lung endothelial cells. How about in aortic endothelial cells?

 Answer:

Because of technical issues and limited access to aortic tissue we decided to isolate mouse lung endothelial cells in order to determine α1AMPK expression. Aortic endothelial cells were not isolated during this project. However, we agree that α1AMPK expression in aortic endothelial cells is of interest and should be investigated in further projects concerning this topic.

  1. In Fig. 2 caption, Mice “with” not “without”?

 Answer:

We changed the caption in figure 2 as suggested.

  1. In Fig. 3A, it looks like that there is a significant difference between the CTR and exercise also for the knockout mice. Please check.

Answer:  

We recalculated the eNOS mRNA expression in the mentioned groups. Although there is a trend visible, it did not reach statistical significance.

  1. Why there is no data for eNOS mRNA for aortic endothelial cells? And no data for the p-eNOS/eNOS for lung endothelial cells?

Answer: 

Thank you very much for this important question. We feel that protein expression in vascular tissue is most relevant to support our functional observations. Therefore not all experiments were repeated with mRNA or in mouse lung endothelial cells.

  1. In Fig. 4C images, are the red spots indicating the ROS production-DHE? More detailed description is suggested for the caption.

Answer:  

We apologize and added necessary details to the caption.

  1. More detailed description is necessary for the caption in Fig. 5.

 Answer:

We added required details to the caption in figure 5.

Discussion

  1. More detailed description should be given for the discrepancy in vascular function between the global and endothelial specific α1AMPK knockout mice.

Answer:  

Thank you very much for highlighting this important issue. The changes in endothelial function between the global and endothelial specific knockout models are particularly different in the response to exercise. While mice with a global deletion of a1AMPK had no improvement of endothelial function, mice with an endothelial specific deletion showed even an impairment of endothelial function. This unexpected finding might relate to the different genetic backgrounds (global a1AMPK deletion was carried out on a FVB background, as C57Bl6 mice with this deletion were not viable) and the high amount of nitrosative stress (peroxynitrite) followed by eNOS uncoupling in endothelial specific a1AMPK deletion. We expanded the discussion part of the manuscript accordingly.

Round 2

Reviewer 2 Report

The authors explain that the low suppression of α1AMPK expression in the KO endothelial cell is due to the low purity of the cells. However, many other papers (including Park et al., EMBO reports, 2016 (PMID: 26755743) and Wang et al., Scientific Reports, 2019 (PMID: 30728372), and Weissman et al., Nature Communications,  2012,) demonstrated the isolation of MLEC with high purity using MACS and the expression of target gene was decreased more than 90%. Therefore, the purity of MLEC should be improved substantially to obtain more reliable results.

Round 3

Reviewer 2 Report

Congratulations! Based on the western and mRNA expression data, the purity of MLEC seems to be improved significantly. I suggest to include the mRNA data in the text or as a supplementary figure. I am wondering if the authors repeated the analysis experiments with newly isolated MELCs and if there is no difference between pure MLECs and impure ones. There are many routes for superoxide to be accumulated in the cell. For expmaple, decrease of SODs and increase of producers such as xanthine oxidase, cytochrome p450, and electron transport system, except NOXs and eNOS. It is better to discuss this point.

In case that the results obtained with newly isolated MLECs are similar with those obtained previously and more discussion on superoxide is added, the revised manuscript is good enough to be published in the 'Antioxidants'.
